# Biosynthesized Poly(3-Hydroxybutyrate) on Coated Pineapple Leaf Fiber Papers for Biodegradable Packaging Application

**DOI:** 10.3390/polym13111733

**Published:** 2021-05-26

**Authors:** Pilanee Vaithanomsat, Kunat Kongsin, Chanaporn Trakunjae, Jirachaya Boonyarit, Amnat Jarerat, Kumar Sudesh, Rungsima Chollakup

**Affiliations:** 1Kasetsart Agricultural and Agro-Industrial Product Improvement Institute (KAPI), Kasetsart University, Bangkok 10900, Thailand; aappln@ku.ac.th (P.V.); kunat.k@ku.th (K.K.); aapcpt@ku.ac.th (C.T.); aapjab@ku.ac.th (J.B.); 2Center for Advanced Studies in Tropical Natural Resources, National Research University-Kasetsart University, Kasetsart University, Bangkok 10900, Thailand; 3Biotechnology of Biopolymers and Bioactive Compounds Special Research Unit, Department of Biotechnology, Faculty of Agro-Industry, Kasetsart University, Bangkok 10900, Thailand; 4Ecobiomaterial Research Laboratory, School of Biological Sciences, Universiti Sains Malaysia, Penang 11800, Malaysia; ksudesh@usm.my; 5Food Technology Program, Kanchanaburi Campus, Mahidol University, Kanchanaburi 71150, Thailand; amnat.jar@mahidol.edu

**Keywords:** poly(3-hydroxybutyrate), pineapple leaf fiber, paper, dip-coating, biodegradable packaging, soil burial test

## Abstract

This paper is aimed at investigating the usage of biosynthesized poly(3-hydroxybutyrate) (P(3-HB)) for a coating on pineapple leaf fiber paper (PLFP). For this purpose, (P(3-HB)) was produced by *Rhodococcus pyridinivorans* BSRT1-1, a highly potential P(3-HB) producing bacterium, with a weight-average molecular weight (*M*_w_) of 6.07 × 10 ^−5^ g/mol. This biosynthesized P(3-HB) at 7.5% (*w/v*) was then coated on PLFP through the dip-coating technique with chloroform used as a solvent. The respective coated PLFP showed that P(3-HB) could be well coated all over on the PLFP surface as confirmed by scanning electron microscopy (SEM) and Fourier transform infrared (FTIR) spectroscopy. The brightness and mechanical properties of PLFP could be improved by coating with biosynthesized P(3-HB) in comparison to commercially available P(3-HB) and non-coated PLFP. Furthermore, coating of P(3-HB) significantly increased the water drop penetration time on the surface of PLFP and was similar to that of the commercial P(3-HB) with the same content. The results showed that all the coated PLPF samples can be degraded under the soil burial test conditions. We have demonstrated that the P(3-HB) coated PLFP paper has the ability to prevent water drop penetration and could undergo biodegradation. Taken together, the P(3-HB) coated PLFP can be applied as a promising biodegradable paper packaging.

## 1. Introduction

Pineapple leaves are important agricultural waste generated from pineapple cultivation, especially in tropical countries such as the Philippines, Brazil, Costa Rica, Thailand and China [1]. It has been documented that pineapple leaf fiber possesses high percentage of cellulose content (70–82%), excellent tensile strength and high toughness, which make them suitable for applying in term of reinforcing composites [2,3,4,5,6]. Besides, pineapple leaf fiber paper could be considered as an outstanding resource for pulp and paper production due to its favorable mechanical properties and high cellulose content [1,7].

In recent years, the increase of consumer awareness on reducing synthetic packaging waste have put new demands on the development of edible/biodegradable packaging from natural materials [8]. Biodegradable paper is also widely developed for food packaging purposes. However, the paper requires a coating process to improve barrier and water protection properties [9]. Petroleum-based derivatives such as polyethylene (PE), ethylene vinyl alcohol (EVOH), polyvinylidene chloride (PVDC), waxes and/or fluorine-derivatives are typically used as coating materials for papers. These synthetic polymers aggravate environmental and economic concerns because of their poor recyclability of coated papers and lack of biodegradation. From a sustainable point of view, several bio-based polymers have become interesting in terms of paper or paperboard coatings such as polysaccharides, proteins, lipids, biodegradable polyesters (poly-hydroxyalkanoate (PHA) and polylactic acid (PLA)) [9,10]. 

Poly(3-hydroxybutyrate) (P(3-HB)) is the most common PHA, which is produced and accumulated intracellularly by various kinds of microorganisms as carbon and energy sources under the nutrient limitation [11]. The P(3-HB) is currently serving as an attractive bioplastic that can be applied in various research fields including polymer blends, nanocomposites, food packaging films and biomedical materials due to being fully biodegradable and biocompatible [12,13]. Additionally, several authors reported that P(3-HB) can be used as a bio-based coating to improve surface hydrophobicity and mechanical properties of papers or paper board [14,15,16]. The coating of P(3-HB) on papers could be carried out by different techniques such as dip coating/solvent casting [16,17], extrusion coating [18] and compression molding [19,20,21]. However, the dip coating is considered to be a facile method of applying an aqueous coating solution over the paper, which is practically applied in the laboratory scale and the coating performance can be quickly determined [10]. We recently reported the production of biosynthesized P(3-HB) by a newly isolated rare actinomyces *Rhodococcus*
*pyridinivorans* BSRT1–1 [22]. However, the application of this newly biosynthesized has not been reported so far.

Therefore, this study aimed to produce P(3-HB) from *Rhodococcus pyridinivorans* BSRT1–1, a rare P(3-HB) producing bacterial strain and utilize as a coating material on PLFP through the dip-coating method with chloroform used as a solvent. The effect of the biosynthesized P(3-HB) coating on physical, mechanical and water absorption properties of coated PLFPs were evaluated and compared to that of the commercial P(3-HB) coating. In addition, biodegradability of biosynthesized P(3-HB) coated PLPF was evaluated by soil burial test.

## 2. Materials and Methods

### 2.1. Materials and Chemicals

A commercial poly(3-hydroxybutyrate) (P(3-HB)), with a weight-average molecular weight (*M*_w_) of 5.50 × 10^5^ g/mol, was supplied by Goodfellow, Cambridge Ltd., Huntingdon, England, UK. Pineapple leaf fibers were produced using a decorticating machine by a group of farmers in Ban Kha, Ratchaburi, Thailand. The obtained pineapple leaf fibers were washed several times with tap water and then dried and stored in a plastic bag at room temperature until used. Chloroform (analytical grade) was provided from VWR Chemicals, Radnor, PA, USA.

### 2.2. Bacterial Strain and Media

*Rhodococcus pyridinivorans* BSRT1–1 was isolated from the soil by Enzyme and Microbial Technology Laboratory, Kasetsart Agricultural and Agro-Industrial Product Improvement Institute (KAPI), Kasetsart University, Bangkok, Thailand, which was previously identified as a high potential P(3-HB) producing bacterium [22]. The minimal medium (MM) was used for P(3-HB) production consisting of NH_4_Cl, 0.5 g/L; KH_2_PO_4_, 2.8 g/L; Na_2_HPO_4_, 3.32 g/L; MgSO_4_·7H_2_O, 0.25 g/L, and 1 ml/L of trace element (TE) solution. The TE solution was comprised of: ZnSO_4_·7H_2_O, 1.3 g/L; FeSO_4_·7H_2_O, 0.2 g/L; (NH_4_)_6_Mo_7_O_24_·4H_2_O, 0.6 g/L; H_3_BO_3_, 0.6 g/L, and CaCl_2_, 0.2 g/L. Fructose and KNO_3_ were used as carbon and nitrogen sources for P(3-HB) production by this strain.

### 2.3. Production and Purification of Poly (3-Hydroxybutyrate) 

Production and purification of P(3-HB) were conducted in accordance with the method described previously [22]. Briefly, *R. pyridinivorans* BSRT1–1 was activated in tryptic soy agar (TSA) plate and incubated at 35 °C for 24 h. The respective colonies were transferred into 250 flask containing 50 mL of in tryptic soy broth (TSB) and incubated in a rotary shaker at 35 °C, 180 rpm for 18 h. The obtained culture was then inoculated into a 10 L stirred tank fermenter containing 6 L of minimal medium (MM) at 35 °C for 48 h with aeration rate at 0.75 vvm, agitation rate at 180 rpm and controlling of pH to neutral within fermentation period. The bacterial cell was harvested after fermentation and then freeze dried at −80 °C. The dried cell was extracted with chloroform at ratio of 1:100 (*w*/*v*) under continuous stirring at ambient temperature for 3 days. The bacterial cell debris was removed by filtration with Whatman No.1. The filtered solution was concentrated using rotary evaporation and followed by slow dropwise into cold method in order to precipitated and purify P(3-HB). The purified P(3-HB) was collected by centrifuge (Kubota 6500, Osaka, Japan) at 10,000× *g*, 4 °C for 10 min and air dried for overnight. 

### 2.4. Characterization of Poly (3-Hydroxybutyrate) 

The weight-average molecular weight (*M*_w_), number-average molecular weight (*M*_n_) and polydispersity index (PDI) of biosynthesized P(3-HB) were analyzed by gel permeation chromatography (GPC) (Agilent 1200 GPC, Santa Clara, CA, USA), following the method that previously published [23]. The P(3-HB) solution was prepared at 1 mg/mL using chloroform as solvent and filtered through 0.22 µm polytetrafluoroethylene (PTFE) membrane before analysis. The 50 µL of filtered P(3-HB) solution was injected into GPC coupled with refractive index detector (RID) equipped with Shodex GPC K-806M and K-802 column at 40 °C when chloroform used as mobile phased at 1 mL/min of flow rate. 

### 2.5. Preparation of Pineapple Leaf Fiber Paper (PLFP)

Pineapple leaf fiber paper (PLFP) was prepared by the slightly modified method previously report [24]. The pineapple leaf fiber was pretreated with 5% (*w*/*v*) of sodium hydroxide solution at 90 °C for 4 h and then allowed to stand at room temperature for 18 h. The alkaline treated fiber was washed several times with tap water until the pH of water reached neutral. The obtained fibers were disintegrated into pulp by a pulping machine (produced by Kasetsart Agricultural and Agro-Industrial Product Improvement Institute (KAPI), Kasetsart University, Bangkok, Thailand) for 25 min. Polyethylene oxide agent was added into pulp and the paper was sheeted using the forming mold. The forming paper was allowed to dry at room temperature around 24 h.

### 2.6. Coating Procedure of P(3-HB) on Pineapple Leaf Fiber Paper

A commercial P(3-HB) (Goodfellow, Cambridge Ltd., Huntingdon, England, UK) was coated on PLFP with different amounts of P(3-HB) (5, 7.5 and 10% (*w*/*v*)) by the dip-coating method as described previously [16]. The P(3-HB) solution was prepared at 60 °C for 6 h using chloroform as a solvent under continuous stirring. The obtained solution was cool down to room temperature and poured into a glass tray and the PLFP was dip-coated into the P(3-HB) solution for 15 min. The resultant PLFPs were dried at room temperature for at least 12 h to remove the residue of the solvent. Due to the limited amount of biosynthesized P(3-HB) samples available, the varying of P(3-HB) concentrations was not carried out. Thus, biosynthesized P(3-HB) was used to coat on PLFP at the optimum content under the same coating condition to commercial P(3-HB). All coated papers were conditioned at 23 °C and 50% relative humidity for at least 8 h prior to testing. The visual appearance of the non-coated and P(3-HB) coted PLFP are presented in Figure 1.

### 2.7. Physical Characterization of P(3-HB) Non-Coated and Coated Pineapple Fiber Paper

#### 2.7.1. Thickness Basis Weight and Percentage of Increasing in Weight

The thickness of paper samples was measured using a digital thickness gage (Mitutoyo, ID-C112XBS, Kanagawa, Japan) with a precision of 0.001 mm at least five random locations of each paper sample. The basis weight (g/m^2^) is referred to the weight of paper per one unit of paper area under specific relative humidity (50% at 23 °C), which calculated by dividing of the weight of paper sample (g) by the area of paper sample (m^2^). The percentage increase in weight was determined following Equation (1):The percentage increase in weight = [(W_t_ − W_0_)/W_0_ × 100](1)
where W_0_ is the initial paper weight and W_t_ is the coated paper weight. 

#### 2.7.2. Color Properties

The colorimetric parameters of paper samples were measured using a portable colorimeter (Hunter Lab Miniscan EZ 4500L, Reston, VA, USA) based on the CIELAB color system. The colorimeter was standardized with standard white and black calibration tiles prior to measuring samples. Measurements were carried out in five replicates at random positions on the PLFP surface. The color values were measured in term of *L** (lightness), *a** (red-green), *b** (yellow-blue) and percentage of brightness followed by the standard method of TAPPI T 452 om-18 [25]. The total color difference (*ΔE**) compared with non-coated PLFP was calculated using equation (2):*ΔE** = [(*L**)^2^ + (*a**)^2^*+* (*b**)^2^]^1/2^(2)

### 2.8. Scanning Electron Microscopy (SEM) of P(3-HB) Non-Coated and Coated Pineapple Fiber Paper

The paper samples were immersed in liquid nitrogen and subsequently cut down by a sharp blade. The surface microstructure of paper samples was observed under scanning electron microscope (SEM) (Hitachi SU8020, Krefeld, Germany) with accelerating voltage of 15 kV. 

### 2.9. Fourier Transform Infrared Spectroscopy (FTIR) of P(3-HB) Non-Coated and Coated Pineapple Fiber Paper

Fourier transform infrared (FTIR) spectrometer (Thermo Scientific Nicolet IR200, Waltham, MA, USA) was used to investigate the chemical functional groups on the surface of paper samples. FTIR spectra were collected over the range of 400–4000 cm^−1^, 128 scans and a resolution of 4 cm^−1^ in attenuated total reflection (ATR) mode.

### 2.10. Mechanical Characterization of P(3-HB) Non-Coated and Coated Pineapple Fiber Paper

#### 2.10.1. Tensile Properties (Tensile Index and Elongation at Break)

The tensile index of paper samples was analyzed in accordance with TAPPI T 494 om-13 [26] Schopper tensile tester (Kumagai Riki Kogyo Co., Ltd., Tokyo, Japan). Paper samples were cut into rectangular strips (15 mm × 150 mm) and then clamped the both ends of each specimen strips with the paper-based holder. The tensile testing was performed at a strain rate of 25 ± 5 mm/min and a clamp distance of 100 mm. The breaking force value (N) was recorded and used to determine the tensile index (N·m/g) according to the following Equation (3):Tensile index (N·m/g) = [653.8 × breaking force]/basis weight(3)

Elongation at break (%) was calculated by dividing the extension at the breakage by the initial gauge length of the samples and multiply by 100.

#### 2.10.2. Folding Endurance 

The folding endurance test was investigate followed by TAPPI T 423 cm-07 [27] using MIT folding endurance tester (Kumagai Riki Kogyo Co., Ltd., Tokyo, Japan). Paper sample were prepared as described in tensile testing. The applied tension was fixed at 1 kg (9.81 N). The folding endurance or the number of double folds is defined as the number of repeated forward and backward folds the specimen, which can withstand under a 1 kg tension before it breaks.

#### 2.10.3. Tear Index

Tearing resistance of paper samples was analyzed by a tearing strength tester (Kumagai Riki Kogyo Co., Ltd., Tokyo, Japan) according to TAPPI T 414 om-12 (Elmendorf method) [28]. Test specimens were prepared in rectangular size of 6.3 cm × 10 cm. The tear resistance force (N) was recorded and then calculated the tear index according to the following Equation (4):Tear index (mN·m^2^/g) = [9.807 × tear resistance]/basis weight(4)

#### 2.10.4. Burst Index

Burst index was determined using a Mullen bursting strength tester (Kumagai Riki Kogyo Co. Ltd., Tokyo, Japan) according to TAPPI T 403 om-15 [29] A rectangular paper sample (12.5 cm × 12.5 cm in size) was clamped firmly between two steel annular plates. A rubber diaphragm under one plate was pressurized by a fluid, causing the diaphragm to bulge. The pressure was increased at a constant rate until the bulging diaphragm caused the paper sheet to rupture. A pressure gauge on the instrument provided a measure of the bursting pressure needed to rupture the paper. Bursting strength was reported in kilopascals (kPa) and then the burst index was calculated using following Equation (5):Burst index (kPa·m^2^/g) = burst strength/basis weight(5)

### 2.11. Soil Burial Biodegradability Test

The soil biodegradation study was performed in laboratory scale following a method described by [30] with some modifications. The paper samples were cut in specimens with the size of 2.5 cm × 2.5 cm and dried in a hot air oven at 50 °C until a constant weight. Three replicates for each sample were buried into a commercial soil at 4–6 cm depth in a plastic box and left at 58 °C for 56 days. The distilled water was added at a certain level into the soil every 2–3 days to ensure the sufficient moisture during the test. The samples were removed from the soil for every 7 days, brushed softly, washed several times with distilled water and then dried at 50 °C until a constant weight. The degree of degradation of paper samples were calculated by normalizing the paper sample weight at different days of incubation respect to the initial value by using following Equation (6):Degree of degradation (%) = [(m_i_ − m_r_)/m_i_] × 100(6)
where m_i_ = the initial weight of the dry sample; m_r_ = the weight of the dry sample after the test.

### 2.12. Water Drop Penetration Test

Water drop penetration test of paper samples were revealed by followed the standard method of TAPPI T 835 om-14 [31] with some modification. One droplet of distilled water (10 µL) was dropped on surface of a square paper samples (2.5 cm × 2.5 cm) using an autopipette and counting the time, in which the drop of water was whole absorbed by the papers.

### 2.13. Statistical Analysis

All experiment data were calculated from at least 9 replicate and expressed as mean ± SD. Analysis of variance (ANOVA) was performed by Duncan’s multiple-range test (DMRT) using the SPSS software (SPSS for Windows, SPSS Inc., Chicago, IL, USA) at *p* ≤ 0.05.

## 3. Results and Discussion 

### 3.1. Characteristics of Biosynthesized P(3-HB)

The *Rhodococcus pyridinivorans* BSRT1–1, a rare actinomyces strain, produced high yield of P(3-HB) (46.8 ± 2.0% based on dry cell weight) in a 10 L bioreactor under the optimum condition previously published by our research group [22]. In this study, gel permeation chromatography (GPC) was used to evaluate the molecular weight of P(3-HB) to understand the general characteristics of the P(3-HB) polymer. Owing to the molecular weight and molecular weight distribution of polymers, it plays a critical role in regards to end-use product properties of polymers including strength, toughness, flow properties, shear viscosity and elasticity [32,33]. It has been demonstrated that the average molecular weight of P(3-HB) depended on several factors such as bacterial strains, medium composition and carbon sources, a fermented condition and the downstream process [34]. The weight (*M*_w_) and number (*M* Hitachi SU8020, Krefeld, Germany) -average molecular weight and respective polydispersity index (PDI) (*M*_w_/*M*_n_) of the P(3-HB) were 6.07 × 10 ^5^ g/mol, 2.96 × 10 ^5^ g/mol and 2, respectively. The PDI reflects the degree of heterogeneity of the polymer’s chain lengths. Hence, the P(3-HB) showed the acceptable heterogenous because the PDI of P(3-HB) produced by wild-type bacteria is usually about 2.0 [35]. Table 1 shows the comparison among the average molecular weight (*M*_w_, *M*_n_) and PDI of the P(3-HB) synthesized in this study and other studies. The average molecular weight of P(3-HB) synthesized in this study resembled the P(3-HB) produced by *R. equi* [23] and *Bacillus subtilis* MSBN17 [36]; all of them had average molecular weight lower than 10 ^6^ g/mol, indicating the acceptable molecular weight of P(3-HB) and qualifying for commercial application [37].

### 3.2. Physical and Color Properties of P(3-HB) Non-Coated and Coted Pineapple Leaf Fiber Papers (PLFP)

The thickness of PLFP did not change with increasing of commercial P(3-HB) concentrations, whereas the weight (g) and basis weight (g/m^2^) of the commercial P(3-HB) coated PLFPs at 7.5 and 10% (*w/v*) were significantly increased (*p* ≤ 0.05). The colorimetric parameters of non-coated and coated PLFP are presented in Table 2. The coating of P(3-HB) significantly improved lightness (*L**) and brightness of PLFP (*p* ≤ 0.05), while the redness (+*a**) and the yellowness (+*b**) of the commercial P(3-HB) coated PLFP was slightly lower than those of the non-coated PLFP (Table 3). By comparing the commercial P(3-HB) to the biosynthesized P(3-HB) on coating of PLFP at 7.5% (*w/v*), the weight (g), basis weight (g/m^2^) and percentage of increasing in weight of both coated PLFP was shown to be non-significantly different. The lightness (*L**) of biosynthesized P(3-HB) coated PLFP was slightly lower, but the brightness was significantly higher than that of the commercial P(3-HB) coated PLFP (Table 3). Moreover, high concentration of commercial P(3-HB) had a slightly increasing effect on the total color difference (*∆E**) of the commercial P(3-HB) coated PLFP, while the biosynthesized P(3-HB) showed the lowest *∆E** compared to the non-coated PLFP (Table 3). This clearly indicated that P(3-HB) synthesized in this study could enhance the brightness of PLFP attributed to the whiteness nature of P(3-HB) polymer. 

### 3.3. Scanning Electron Microscopy (SEM)

The SEM micrograph of the surface and cross section of the non-coated PLFP and the commercial or P(3-HB) coated PLFPs are illustrated in Figure 2. The non-coated PLFP showed the disordered entanglements of pineapple fibers network to have many pores, whereas the coted PLFPs (with commercial or biosynthesized P(3-HB)) showed a smooth layer of P(3-HB) covering the entire surface of PLFPs with no visible pores, which was clearly contrast with the fibrous network of the non-coated PLFP. It was observed that some amount of P(3-HB) embedded to the pineapple fibers on surface of PLFPs. For cross section, the commercial P(3-HB) and P(3-HB) coated PLFP showed a compact structure of P(3-HB) on top surface of PLFPs, while the non-coated PLFP showed a fibrous layer. This result confirmed that both the commercial and biosynthesized P(3-HB) could be well coated all over the PLFP surface via the dip-coating techniques.

### 3.4. Fourier Transform Infrared Spectroscopy (FTIR)

Chemical structure of P(3-HB) non-coated and coated PLFPs were studied by Fourier transform infrared spectroscopy (FTIR), and the FTIR spectra are shown in Figure 3. The spectra of PLFP showed the peak at 3380 cm ^−1^ which is attributed to the absorption of hydroxyl groups (OH-). While the commercial and biosynthesized P(3-HB) showed the characteristic bands located at 1718–1720 cm ^−1^ assigned to the ester carbonyl groups (C=O), the band at 1043–1054 cm^−1^ was assigned to C-O ester bond stretching. Other bands located at 1268–1282 cm^−1^ were assigned to methylene group (-CH_2_) and the band at 1382–1386 cm^−1^ was associated with methyl group (-CH_3_) [15]. The spectra of P(3-HB) non-coated and coated PLFPs were similar to the characteristic of P(3-HB) with a slight reduction of hydroxyl group at the band of 3380 cm^−1^. The FTIR result indicated that the film of commercial and biosynthesized P(3-HB) was well generated by the dip-coating technique over the entire surface of PLFP.

### 3.5. Mechanical Properties

The effect of commercial P(3-HB) contents (5, 7.5 and 10% (*w*/*v*)) on mechanical properties of coated PLFP is shown in Figure 4. It was found that coating of commercial P(3-HB) did not affect the tensile index of PLFP. The percentage of elongation, tear index and burst index of the commercial P(3-HB) coated PLFP were significantly reduced. As it is well known that P(3-HB) is a brittle biopolymer due to its high degree of crystallinity, this result might therefore be influenced by the matrix of commercial P(3-HB) retained in the internal porous space of PLFPs after the coating process which resulted in an increased brittleness of the coated papers [41]. On the contrary, the folding endurance of commercial P(3-HB) coated PLFP was significantly increased compared with the non-coated PLFP (*p* ≤ 0.05). This increase of folding endurance might be associated with the extra coating weight on the surface of the coated paper which resulted in the resistance to the repeated folding test [42]. The biosynthesized P(3-HB) coated PLFP at 7.5% (*w*/*v*) was significantly higher in tensile index, percentage of elongation, folding endurance and burst index than that of the commercial P(3-HB) coated PLFP at all P(3-HB) concentrations and the non-coated PLFP (*p* ≤ 0.05). Tear index of the biosynthesized P(3-HB) coated PLFP was similar to the commercial P(3-HB) coated PLFP at 5 and 7.5% (*w*/*v*). It is remarkable that both the commercial P(3-HB) and biosynthesized P(3-HB) coated papers showed lower tear index than that of the non-coated paper. This might be due to the matrix of P(3-HB) which decreased the inter-fiber bonding of papers and resulted in the decreased tear index of coated papers [43]. An improvement on mechanical properties of the P(3-HB) coated papers might be explained in term of the difference of average molecular weight between the commercial and biosynthesized P(3-HB).

### 3.6. Water Drop Penetration 

Water drop penetration test of non-coated and coted PLFPs were performed by the detection of water drop penetration time (second) on the surface of non-coated and coated PLFPs (Figure 5.). The result showed that the water drop was suddenly absorbed by the non-coated PLFP, which then resulted in the lowest water drop penetration time (0 s). Water drop penetration time of the commercial P(3-HB) coated PLFPs were significantly increased when compared to the non-coated PLFP, especially for high commercial P(3-HB) contents at 7.5 and 10% (*w/v*), which were higher than that of non-coated PLFP for 18 and 26 times, respectively. In addition, the water drop penetration time of biosynthesized P(3-HB) coated PLFP was close to that of the commercial P(3-HB) coated PLFP. This result clearly demonstrated that hydrophobic P(3-HB) could delay water absorption on PLFP surfaces.

### 3.7. Soil Burial Biodegradability 

The coated PLFPs, having predominant mechanical properties and acceptable water drop penetration time, were selected to evaluate and compare the biodegradability. The biosynthesized P(3-HB) coated PLFP with varying contents (0%, 5%, 7.5% and 10% (*w/v*)) were buried into the moist soil for a period of 56 days at 63 °C. The visual appearance of degraded paper samples recovered at different testing times are presented in Figure 6. It was observed that the residual samples at 49 and 56 days were darker and more fragile, as could be seen in non-coated PLFP and PLFP coated with 5% and 7.5% (*w/v*) of P(3-HB). In addition, it was noticeable that increasing of P(3-HB) content to 10% (*w/v*) decelerated the degree of degradation of the coated paper (42% after 56 days), while the non-coated and P(3-HB) coated PLFP at 5% and 7.5% (*w/v*) showed higher degree of degradation around 53–60% after 56 days (Figure 7.). This result might be due to the high amount of crystalline P(3-HB), which slowed down the degradation phenomenon [44]. However, the rate of biodegradation of biocomposites depends on many environmental factors such as moisture, light (radiation), temperature and microorganisms [45]. Among these factors, P(3-HB) degradation is mainly enzymatically degraded by various kinds of microorganisms existing in natural soils [46]. Thus, the microbial communities in soil, which are responsible for biodegradation of the P(3-HB) coated PLFP should be evaluated for further study. 

## 4. Conclusions

This study applied the utilization of the poly(3-hydroxybutyrate) (P(3-HB)) produced by *R. pyridinivorans* BSRT1-1, a highly potential P(3-HB) producing bacterium, for dip-coating on pineapple leaf fiber paper (PLFP). It resulted in an improvement of brightness and mechanical properties (tensile index, percentage of elongation, folding endurance and burst index) of PLFPs over that of a commercial P(3-HB) coated and non-coated PLFP. SEM and FTIR studies confirmed that P(3-HB) completely covered the surface of PLFP. This P(3-HB) coated PLFP showed an increase of water drop penetration time compared to the non-coated PLFP, indicating that the P(3-HB) could reduce the water susceptibility of the cellulose of PLFP. Moreover, soil burial biodegradation of the P(3-HB) coated PLFPs occurred rapidly within 56 days. According to these results, the biosynthesized P(3-HB) has the potential as an eco-friendly material for paper coating application to replace the non-renewable polymers.

## Figures and Tables

**Figure 1 polymers-13-01733-f001:**
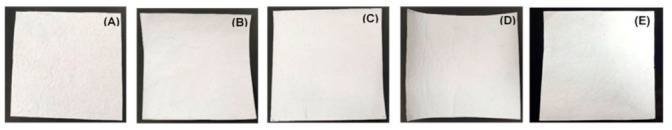
Visual appearance of (**A**) the non-coated PLFP, (**B**–**D**) the commercial P(3-HB) coated PLFPs at 5, 7.5 and 10% (*w/v*) and (**E**) the biosynthesized P(3-HB) coated PLFP at 7.5% (*w*/*v*).

**Figure 2 polymers-13-01733-f002:**
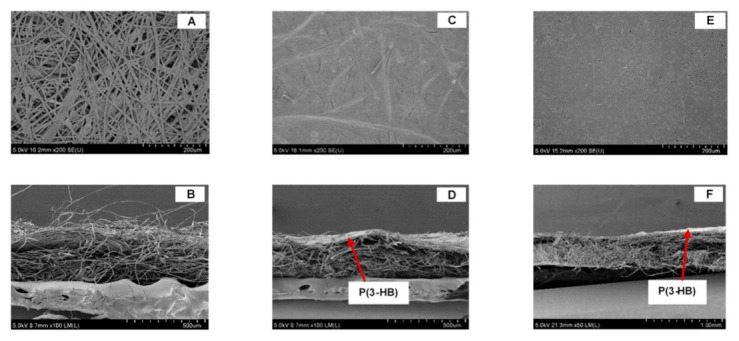
SEM micrographs of the surface and cross section of (**A**–**B**) the non-coated PLFP, (**C**–**D**) the commercial P(3-HB) coated PLFP at 7.5% (*w/v*) and (**E**–**F**) the biosynthesized P(3-HB) coated PLFP at 7.5% (*w/v*).

**Figure 3 polymers-13-01733-f003:**
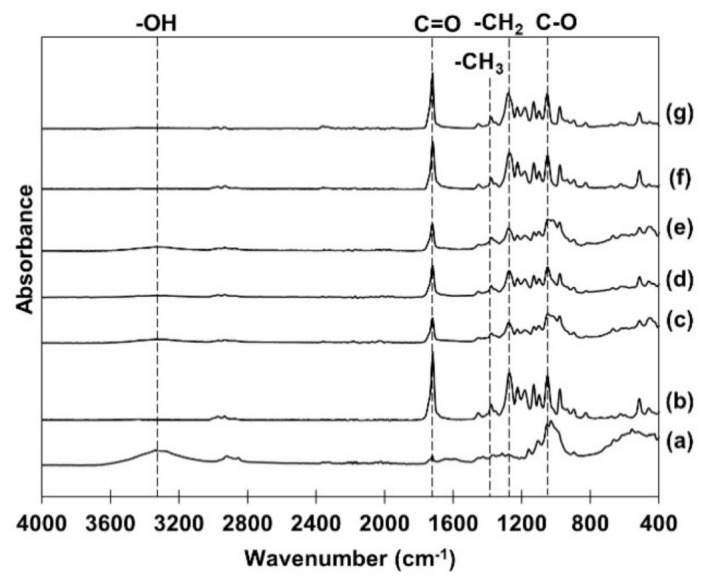
FTIR spectra of (**a**) non-coated PLFP, (**b**) commercial P(3-HB) film, (**c**–**e**) commercial P(3-HB) coated PLFPs at 5%, 7.5% and 10% (*w/v*), (**f**) biosynthesized P(3-HB) film and (**g**) biosynthesized P(3-HB) coated PLFP at 7.5% (*w/v*).

**Figure 4 polymers-13-01733-f004:**
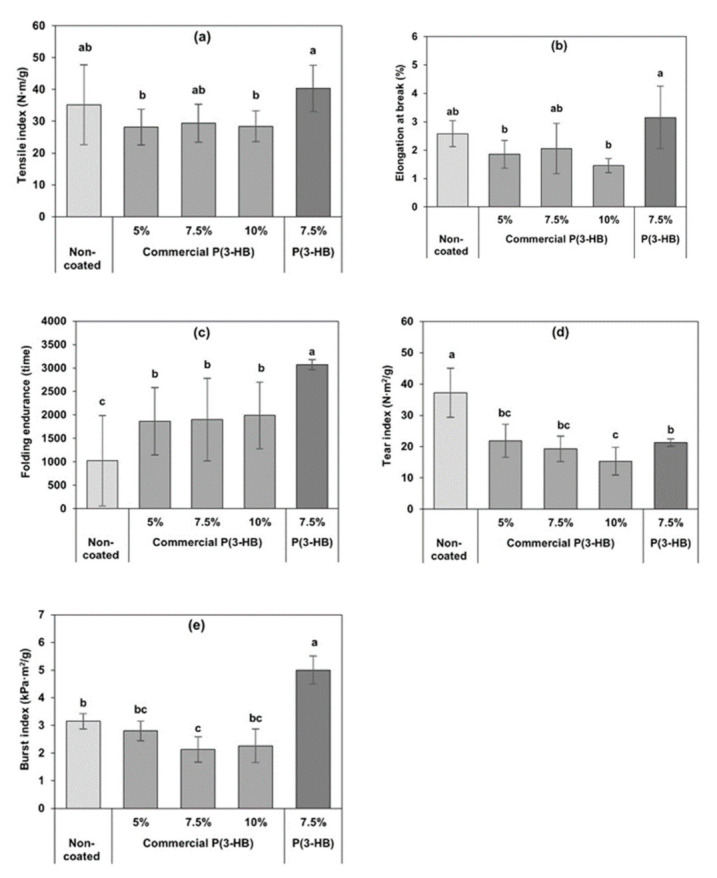
Mechanical properties of the non-coated PLFP, the commercial P(3-HB) coated PLFPs at 5%, 7.5% and 10% (*w*/*v*) and the biosynthesized P(3-HB) coated PLFP at 7.5% (*w*/*v*) including (**a**) tensile index, (**b**) elongation, (**c**) folding endurance, (**d**) tear index and (**e**) burst index. Different letters above graph bars re represent the different between the averages of each parameter at significance level of *p* ≤ 0.05.

**Figure 5 polymers-13-01733-f005:**
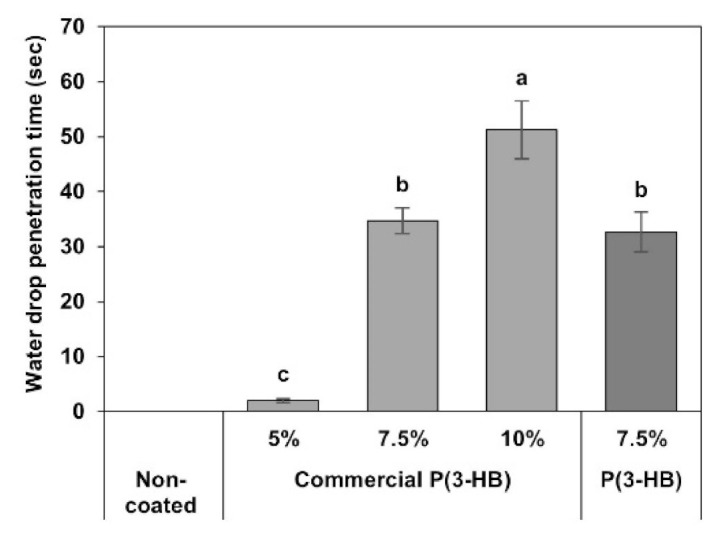
Water drop penetration time on surface of the non-coated PLFP, the commercial P(3-HB) coated PLFPs at 5%, 7.5% and 10% (*w/v*) and the biosynthesized P(3HB) coated PLFP at 7.5% (*w/v*). Different letters above graph bars are represent the different between the averages of each parameter at significance level of *p* ≤ 0.05.

**Figure 6 polymers-13-01733-f006:**
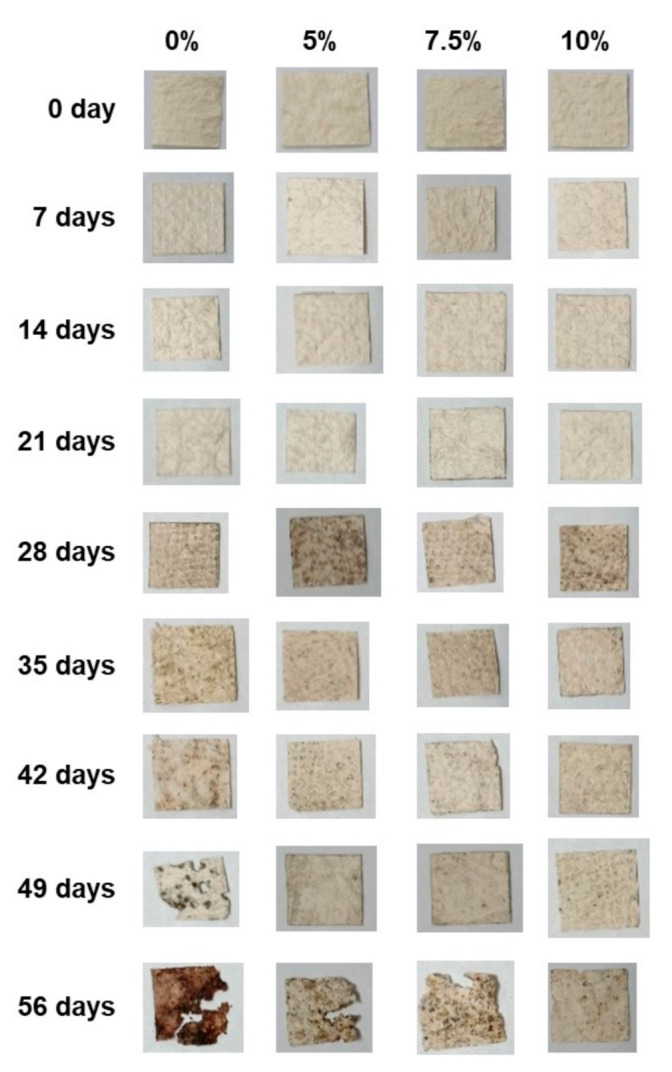
Visual appearance of non-coated and biosynthesized P(3-HB) coated PLFPs (5%, 7.5% and 10% (*w*/*v*)) at different times after burial in soil.

**Figure 7 polymers-13-01733-f007:**
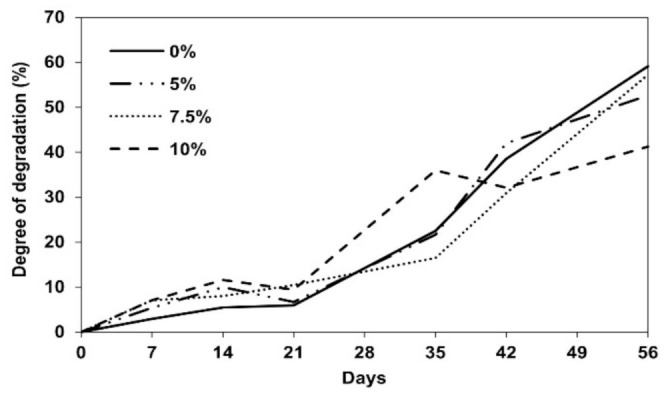
Degree of degradation of non-coated and biosynthesized P(3-HB) coated PLFPs (5%, 7.5% and 10% (*w*/*v*)) at different days of soil burial biodegradability test.

**Table 1 polymers-13-01733-t001:** The weight (*M*_w_) and number (*M*_n_) -average molecular weight and respective polydispersity index (PDI) (*M*_w_/*M*_n_) of the P(3-HB) produced by *R. pyridinivorans* BSRT-1, and comparison with other reports.

Polymer/Carbon Source	Bacterial Strains	*M*_w_ (g/mol)	*M*_n_ (g/mol)	PDI (*M*_w_/*M*_n_)	References
P(3-HB) (fructose)	*R. pyridinivorans* BSRT1-1	6.07 × 10^5^	2.96 × 10^5^	2.0	This study
Commercial P(3-HB)	-	5.05 × 10^5^	-	-	This study
PHB (crude palm kernel oil)	*R. equi*	6.42 × 10^5^	3.73 × 10^5^	1.72	[23]
PHB (soy)	*Cupriavidis necator*	7.90 × 10^5^	3.49 × 10^5^	2.26	[38]
PHB (molasses and corn steep liquor)	*Bacillus megaterium* ATCC 6748	3.90 × 10^6^	2.65 × 10^6^	1.47	[39]
PHB (pulp industry waste)	*Bacillus* *subtilis* MSBN17	6.40 × 10^5^	3.80 × 10^5^	1.68	[36]
P(3HB-*co*-3HV) (bagasse extract)	*Halomonas campisalis*	1.39 × 10^6^	8.39 × 10^5^	1.66	[40]

**Table 2 polymers-13-01733-t002:** Physical properties of the non-coated PLFP, the commercial P(3-HB) coated PLFPs at 5, 7.5 and 10% (*w*/*v*) and the biosynthesized P(3-HB) coated PLFP at 7.5% (*w*/*v*).

Properties	Non-Coated PLFP	Commercial P(3-HB)			Biosynthesized P(3-HB)
		5%	7.5%	10%	7.5%
Weight (g)	1.55 ± 0.19 ^b^	1.79 ± 0.26 ^b^	2.35 ± 0.50 ^a^	2.62 ± 0.47 ^a^	2.50 ± 0.06 ^a^
Basic weight (g/m^2^)	99.64 ± 12.73 ^b^	114.35 ± 16.56 ^b^	150.09 ± 31.88 ^a^	171.30 ± 30.02 ^a^	159.82 ± 3.73 ^a^
The percentage increase in weight (%)	-	60.81 ± 16.04 ^b^	64.69 ± 16.17 ^b^	98.78 ± 12.13 ^a^	59.62 ± 7.64 ^b^
Thickness (mm)	0.27 ± 0.05 ^b^	0.29 ± 0.05 ^b^	0.29 ± 0.06 ^b^	0.27 ± 0.03 ^b^	0.59 ± 0.01 ^a^

Different letters (a, b) in the same row mean that the results are significantly different at *p ≤* 0.05, while ns means that the results are not significantly different at *p* > 0.05 by Duncan’s multiple-range test.

**Table 3 polymers-13-01733-t003:** Colorimetric parameters of the non-coated PLFP, the commercial P(3-HB) coated PLFPs at 5, 7.5 and 10% (*w/v*) and the biosynthesized P(3-HB) coated PLFP at 7.5% (*w/v*).

Color Parameters	Non-Coated PLFP	Commercial P(3-HB)			Biosynthesized P(3-HB)
		5%	7.5%	10%	7.5%
*L**	87.92 ± 0.89 ^b^	91.22 ± 0.35 ^a^	91.23 ± 0.33 ^a^	91.23 ± 0.39 ^a^	87.63 ± 0.22 ^b^
*a**	1.23 ± 0.04 ^a^	0.86 ± 0.15 ^c^	1.00 ± 0.12 ^b^	0.86 ± 0.06 ^c^	1.01 ± 0.02 ^b^
*b**	9.21 ± 0.09 ^a^	6.91 ± 0.28 ^b^	7.17 ± 0.47 ^b^	7.11 ± 0.39 ^b^	8.16 ± 0.02 ^b^
*∆E**	-	4.03 ± 0.07 ^b^	6.14 ± 0.17 ^a^	6.32 ± 0.36 ^a^	1.03 ± 0.25 ^c^
Brightness (%)	27.39 ± 0.60 ^c^	41.98 ± 13.40 ^b^	44.06 ± 2.38 ^b^	45.13 ± 1.59 ^b^	62.57 ± 0.60 ^a^

Different letters (a, b, c) in the same row mean that the results are significantly different at *p* ≤ 0.05 by Duncan’s multiple-range test.

## Data Availability

Data is contained within the article.

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
