# Peer review of "Biosynthesized Poly(3-Hydroxybutyrate) on Coated Pineapple Leaf Fiber Papers for Biodegradable Packaging Application"

_polymers, 2021, doi:10.3390/polym13111733_

Round 1

Reviewer 1 Report

My comments are the following:

  1. The aim of the study is not clearly defined in the abstract.
  2. The introduction part should contain more references containing the possibility to use and produced edible/biodegradable packaging. The following reference should be used: Simona, J., Dani, D., Petr, S., Marcela, N., Jakub, T., & Bohuslava, T. (2021). Edible films from carrageenan/orange essential oil/trehalose—structure, optical properties, and antimicrobial activity. Polymers13(3), 332.
  3. It would be good to include the picture of produced PFLP.
  4. Principal component analysis should be included too.

Reviewer 2 Report

This paper reports on the “Biosynthesized Poly(3-hydroxybutyrate) on Coated Pineapple Leaf Fiber Papers for Biodegradable Packaging Applicatio”. The article is interesting and clearly presented. Methodology, results, discussion and referenceseems to be correct.

I have few comments to the manuscript:

  1. Page 3 line 98. Deleted extra space.
  2. Missing references in methodology.
  3. Page 8 line 286. Deleted extra space.

Taking into account all comments the manuscript may be published in Polymers after minor revision.

Round 2

Reviewer 1 Report

The manuscript can be accepted.